# Adversarial Remote Sensing Scene Classification Based on Lie Group Feature Learning

Chengjun Xu [1,2,*], Jingqian Shu [1] and Guobin Zhu [2]

1   School of Software, Jiangxi Normal University, Nanchang 330022, China
2   School of Remote Sensing and Information Engineering, Wuhan University, Wuhan 430000, China
*   Correspondence: 2018102160001@whu.edu.cn

**Abstract:** Convolutional Neural Networks have been widely used in remote sensing scene classification. Since this kind of model needs a large number of training samples containing data category information, a Generative Adversarial Network (GAN) is usually used to address the problem of lack of samples. However, GAN mainly generates scene data samples that do not contain category information. To address this problem, a novel supervised adversarial Lie Group feature learning network is proposed. In the case of limited data samples, the model can effectively generate data samples with category information. There are two main differences between our method and the traditional GAN. First, our model takes category information and data samples as the input of the model and optimizes the constraint of category information in the loss function, so that data samples containing category information can be generated. Secondly, the object scale sample generation strategy is introduced, which can generate data samples of different scales and ensure that the generated data samples contain richer feature information. After large-scale experiments on two publicly available and challenging datasets, it is found that our method can achieve better scene classification accuracy even with limited data samples.

**Keywords:** generative adversarial network; Lie group; remote sensing; scene classification

## 1. Introduction

High-Resolution Remote Sensing Images (HRRSI) can represent the geometric structure and texture information of ground objects more accurately and clearly [1–3], and also provide us with more accurate earth observation [4–6]. Remote sensing scene classification is one of the most direct ways to understand remote sensing scenes. It marks HRRSI as predefined semantic categories [7–9].

According to the characteristics of different levels, there are three main approaches [2]: (1) Low-level features; (2) Middle-level features; (3) High-level features. Low-level features mainly adopt shape, texture, spectral features, Global Color Histogram (GCH) [10,11], etc. These models are characterized by invariance to image rotation and translation [12], but their classification result is low for complex scenes [8]. Subsequently, a second approach has emerged. This approach focuses on encoding a dictionary of low-level features and can describe details more powerfully [13]. The above two scene classification models are based on manual features and cannot be applied to complex ground object scenes [14]. Compared with the first two manual features, high-level features automatically learn feature information through Convolutional Neural Networks (CNN) [15] and Fine-tuned CNN [16], which can achieve relatively high classification accuracy.

However, to achieve high classification accuracy, deep neural networks require a large number of training sample sets. In fact, the training sample set requires the prior knowledge of experts, and the cost of sample labeling is high. Currently, prior knowledge and experience are mainly used to solve this problem [17], such as transfer learning [18], meta-learning [19], and metric learning [20]. The above methods depend on the feature

representation ability of the model, which is not intuitive and more complex [14]. In the implementation process of the actual algorithm, the data augmentation method is usually used to increase the training samples, such as pixel change, image rotation, and geometric transformation. The training samples obtained by this method are limited in diversity and quantity. Therefore, the sample generation approach is one of the effective approaches to address the above problems, which can effectively increase the number of samples and obtain rich and diverse data samples.

Generative Adversarial Networks (GAN) are one of the typical models for sample generation [14]. This model automatically learns the distribution of data samples through the competition between generator and discriminator and generates a dataset similar to the original sample. For example, Song et al. [21] proposed a spatiotemporal fusion method of remote sensing images based on a generative adversarial network, which is used to process one Landsat-MODIS prior image pair case (OPGAN). Chen et al. [9] proposed a novel model based on remote sensing road-scene neighborhood probability enhancement and improved conditional generative adversarial network, which consists of two parts: road scenes classification and fine-road segmentation section. Lin et al. [22] proposed a multi-layer feature matching Generative Adversarial Network to improve the accuracy of scene classification. Yu et al. [23] proposed a GAN model integrating an attention mechanism to improve accuracy. However, these models are unsupervised models, and the generated samples do not contain labeled information, so they cannot be used to represent specific categories of scenes. Later, Ma et al. [24] proposed the sifting GAN model to enhance the authenticity of the generated samples. However, this model mainly extracts global high-level semantic features, ignoring the correlation between spatial information and features [25,26].

In this study, we proposed a novel supervised adversarial Lie Group feature learning network. The network adopts a supervised model, which can ensure that the generated samples contain category information, and extract more discriminative features such as external physical structure features and internal socio-economic semantic features through Lie Group machine learning.

The main contributions of the study are as follows:

1.  To address the problem of limited data samples, especially when the traditional GAN cannot generate data samples containing scene category information, we propose a novel supervised adversarial Lie Group feature learning network. The model adopts the supervision mode, which inputs the category information and data samples into the generator and discriminator at the same time and optimizes the supervisory anti-loss function to generate samples of category information. In addition, based on the previous feature learning, we added the internal socio-economic semantic features of the scene to further improve the representation ability of the scene model;

2.  To make the generated data samples contain richer semantic feature information, we design the object scale sample generation strategy. This strategy can obtain data samples of different scales and semantic feature information of different scales, and ensure that a single fake sample has more detailed and richer semantic feature information. In addition, the model can effectively suppress the problem of model collapse during training;

3.  To verify the feasibility and effectiveness of our model, we have carried out a large number of experiments. The experimental results show that, compared with other models (including classic methods and state-of-the-art methods), our method can effectively generate data samples, and these data samples contain scene category information. The generated data samples have richer and more detailed semantic feature information, and the samples have diversity.

## 2. Method

To solve the problem of low performance of deep network model in the case of limited data samples, we propose a novel supervised adversarial Lie Group feature learning

network. This model can effectively generate data samples with category information. There are two main differences between our method and traditional GAN: (1) Our model takes category information and data samples as the input of the model, optimizes the constraint of category information in the loss function, and generates data samples containing category information; (2) The introduction of object scale sample generation strategy can generate data samples of different scales, and ensure that the generated data samples contain richer feature information.

Our proposed model is shown in Figure 1. Firstly, the data samples and corresponding category information are fused as the input of the model. Then, Lie Group feature learning is performed on the data, which mainly learns the external physical structure features, internal socio-economic semantic features, and high-level semantic features of the scene. After that, the learned features and corresponding category information are fused into the network, and the data samples containing fake category information are gradually produced from small scale to large scale. Finally, after iterative training, the generated data samples with fake category information were fused with the original category data samples.

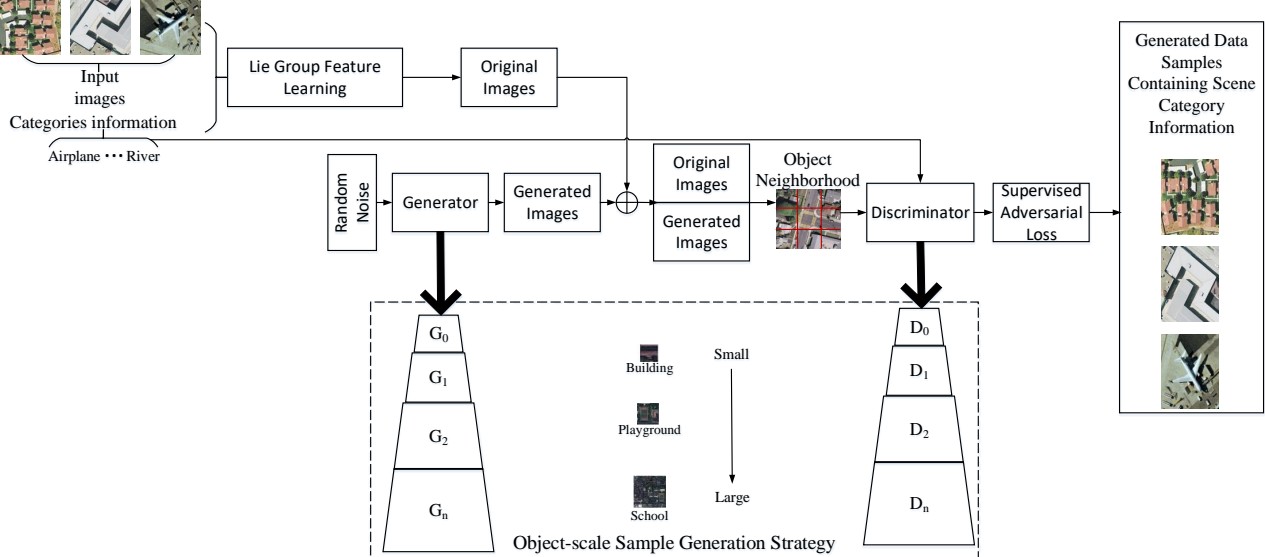

**Figure 1.** Architecture of our proposed model.

## 2.1. Lie Group Feature Learning

In the previous study [2,15], we adopted Lie Group machine learning to select, extract and learn features of sample data, and then constructed the Lie Group region covariance feature matrix to characterize features and the relationship between features. However, we found that the model has difficulty distinguishing remote sensing scenes with the same or similar geometric structure and spatial layout. To address this problem, in the process of feature learning, we supplement the internal socio-economic semantic features of the scene. In other words, compared with previous studies, the main difference is that our proposed model supplements the internal socio-economic semantic features of the scene.

In the actual algorithm model, the internal socio-economic semantic features of an object in the scene are mainly extracted from the Amap and are crawled according to the main categories. To avoid ambiguity effectively, we fuse intermediate categories and subcategories into point semantic objects, such as museums, libraries, archives, universities, secondary schools, primary school, and kindergartens were applied in this study, instead of science. A total of 45 categories are used in this study, and the category information can be added or deleted according to the needs of actual scenarios.

### 2.2. Supervised Condition Generation

Traditional GAN is mainly based on unsupervised generative models whose inputs are out-of-order and unlabeled, learning the characteristics of these data samples. After the model is trained, traditional GAN generates fake data samples that do not contain scene category information. However, such samples are of little significance to remote sensing research.

To address the above problems, the Conditional Generative Adversarial Network (CGAN) model proposed by Mirza and Osindero [27] is used for reference, which can generate data samples with category information. As shown in Figure 1, our model is transformed from unsupervised to supervised. In other words, in the model design, the scene category information and corresponding data samples are taken as the input of the model. In the condition generation module, random noise and condition information form a joint hidden layer.

In the discriminator module, the original datasets and the generated datasets are merged in the channel dimension, and the corresponding category information is input into the model. Compared with the traditional discriminator, our proposed discriminator still needs to make additional judgments: (1) whether the generated data set matches the original data set; and (2) whether the generated data set is valid and authentic.

### 2.3. Object-Scale Sample Generation Strategy

In this subsection, we proposed the object-scale sample generation strategy. The generator and discriminator are extracted according to the scale size, and gradually generate data samples from small scale to large scale.

#### 2.3.1. Generator

As shown in Figure 2, the random noise $rn$ and the category information $ci$ are fused into tensors and used together as the input of the network. In our previous study [2], we found that the ordinary convolution receptive field is relatively small and the number of parameters is large, as shown in Table 1. Therefore, we adopt parallel dilated convolution to expand the receptive field and learn semantically sensitive transformations.

Generally, the rectified linear units (ReLU) activation function is used in the traditional module. However, as shown in Figure 3, the ReLU activation function is directly reduced to zero in the negative semi-axis region, which may lead to the disappearance of potential gradients in the training process of the model, making the generator unable to generate effective image samples and affecting the accuracy of the model. Therefore, we adopted the scaled exponential linear units (SeLU) activation function. The biggest difference between this activation function and the ReLU activation function is that there are negative value existing in the negative semi-axis area.

The mathematical expression is as follows:

$$f(x) = \begin{cases} \alpha x, & x > 0 \\ \alpha(\beta e^x - \beta), & x \leq 0 \end{cases} \tag{1}$$

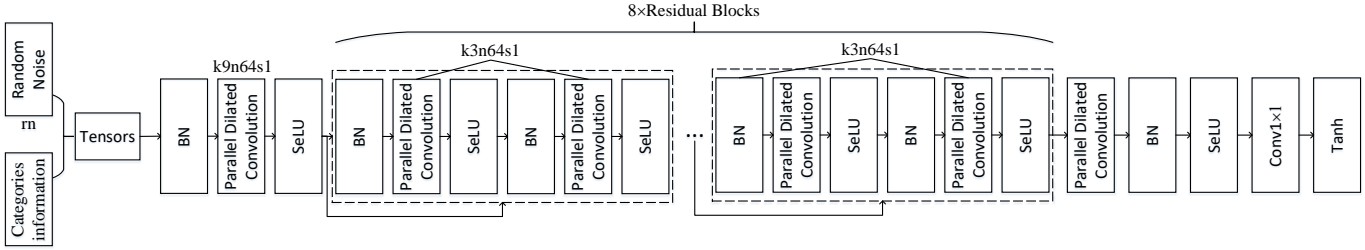

**Figure 2.** Network structure of the generator. k, n, and s indicate kernel size, number of feature maps, and stride, respectively.

**Table 1.** Comparison between different convolutions.

| Methods | Kernel Size | Input Channel | Output Channel | Layer | Parameters Size | Total (M) |
|---|---|---|---|---|---|---|
| Ordinary | $3 \times 3$ | 1024 | 1024 | Conv1 | $1024 \times 1024 \times 3 \times 3 = 9,437,184$ | $23,811,552 \approx 28.3$ |
| | | | | Conv2 | $1024 \times 1024 \times 3 \times 3 = 9,437,184$ | |
| | | | | Conv3 | $1024 \times 1024 \times 3 \times 3 = 9,437,184$ | |
| | $5 \times 5$ | 1024 | 1024 | Conv1 | $1024 \times 1024 \times 5 \times 5 = 26,214,400$ | $78,643,200 \approx 78.6$ |
| | | | | Conv2 | $1024 \times 1024 \times 5 \times 5 = 26,214,400$ | |
| | | | | Conv3 | $1024 \times 1024 \times 5 \times 5 = 26,214,400$ | |
| Parrallel | $5 \times 5$ | 512 | 512 | Conv1 | | $6,553,600 \approx 6.55$ |
| | | | | Conv2 | $512 \times 512 \times 5 \times 5 = 6,553,600$ | |
| | | | | Conv3 | | |

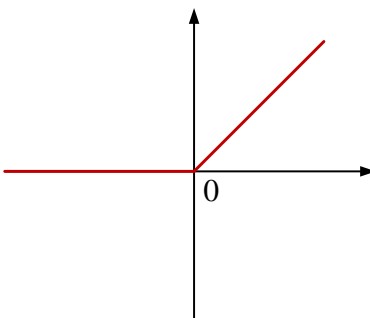

**Figure 3.** RELU activation function.

The main advantages of adopting the SeLU activation function instead of the traditional ReLU activation function are as follows: (1) The saturation zone can effectively suppress the large variances of the lower level; (2) The parameters in the function make the slope of some areas greater than 1, which can effectively adjust the relatively small low-level variance; (3) The output value can effectively control the mean value. After repeated experiments and analysis of the results, the above two hyperparameters $\alpha$ and $\beta$ are set to $\alpha = 1.31$ and $\beta = 1.09$. In the optimized generator module, the mode collapses and gradient disappearance can be effectively suppressed, which makes the generated dataset more authentic and diversified.

In the previous studies, we found that model deepening may lead to model degradation [2]. Therefore, we adopted a residual network with eight residual blocks. As shown in Figure 2, the layout of each residual block is the same, containing two parallel dilated convolutional layers, two Batch Normalization (BN) layers, and two SeLU activation functions. To make the model training more effective and the feature information extracted by the model more richer, the residual block adopts a shortcut (skip connection) to add the input from the upper layer to the lower layer.

After a certain extent of training in the above structures with different resolutions, we continue to adopt parallel dilated convolution, BN, SeLU, and convolutional layer operations to enable the model to effectively learn detailed feature information. In the last layer of the generator module, the Lie Group Tanh activation function is adopted [25,26].

In addition, in the training process of the generator, the number of parameters of each convolution layer is fixed at the previous resolution, and the number of parameters of the parallel dilated convolution is much reduced compared with that of the ordinary convolution. Therefore, the calculation performance of the whole training process of the generator is relatively better, and the calculation amount of parameters is relatively small. Through the above operation, fake data samples of different scales can be generated, and the samples can also contain richer and more detailed information.

### 2.3.2. Discriminator

As shown in Figure 4, in the discriminator module, data samples and category information are taken as the input of the discriminator module together, and the corresponding parallel dilated convolution and SeLU activation functions are adopted. To enhance the classification ability of the discriminator, eight parallel dilated convolution is used to extract the features of the data samples. After two dense layers, the Lie Group Sigmoid activation function outputs the result. The traditional Sigmoid activation function is mainly suitable for vector data samples in Euclidean space. However, the method we proposed is to operate on the Lie Group manifold space, which does not belong to Euclidean space. Moreover, we used matrix data sample representation, and the matrix calculation does not satisfy the commutative law. Therefore, we adopted the previously proposed Lie Group Sigmoid and Lie Group Tanh activation functions. For details, please refer to the literature [25,26].

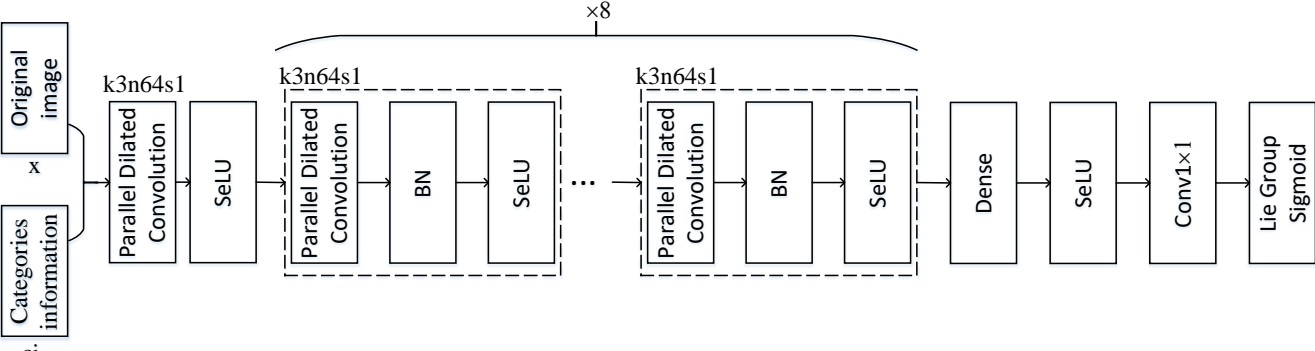

**Figure 4.** Network structure of the discriminator. K, n, and s indicate kernel size, number of feature maps, and stride, respectively.

The generators and discriminators mentioned above are synchronized and mirror each other, so the above strategy can effectively suppress the collapse of the GAN model. In the training process, all modules in the generator and discriminator can be trained, and new modules can be smoothly added to the model, effectively avoiding the impact on the lower resolution layer. The low-resolution data samples generated by the generator are also relatively stable, which effectively reduces the number of samples processed by the discriminator. Therefore, the problem of model collapse during training can be effectively suppressed.

### 2.4. Probability Enhancement Strategy in Ground Object Neighborhood

Neighborhood information refers to the feature information of adjacent ground objects, which is mainly divided into four and eight neighborhoods and is used to represent the neighborhood feature information around the pixels. The spatial correlation of scene objects is usually regarded as the consistency of the connectivity of regional scenes. Inspired by this feature information, we extend the pixel neighborhood theory to remote sensing scenes. Therefore, we design and develop a probability enhancement strategy based on neighborhood correlation to improve the accuracy of sample scenes.

In the traditional model without data samples of geospatial coordinate information, it is generally impossible to construct ground object neighborhood information. However, the regional images neighborhood spatial coordinate information, so the ground object domain information can be constructed, as shown in Figure 5.

As shown in Figure 5, the four neighborhood calculates four adjacent regions, namely, up, bottom, left, and right, while the eight neighborhood has a larger calculation range, namely eight adjacent regions. Ground object scene is a small part in the middle, which is likely to be missed, resulting in missing or wrong scene category information. In this study,

the eight neighborhood approach is adopted to enhance the correlation between ground objects. The mathematical expression is as follows:

$$P = p_i + (6/8) \times p_i \tag{2}$$

where $p_i$ represents the probability of the $i^{th}$ region.

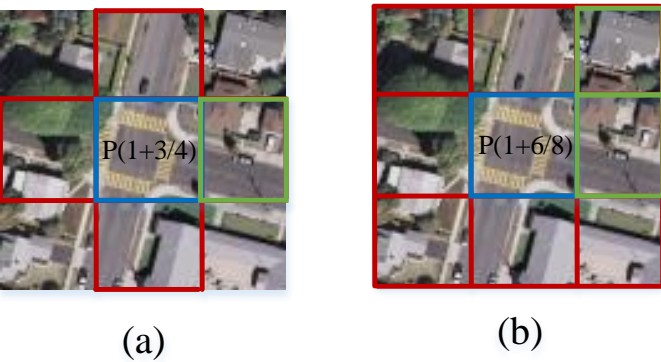

(a)           (b)

**Figure 5.** The expression of neighborhood enhancement of remote sensing objects: (**a**) represents four neighborhoods; (**b**) represents eight neighborhoods, P represents probability.

*2.5. Supervised Adversarial Loss*

Our approach can generate data samples containing category information, and the corresponding loss function is:

$$L_{G_n} = -E_{x_n^f} log(D(x_n^f|ci)) \tag{3}$$

where $D(x_n^f)$ represents the final output of the discriminator, $x_n^f$ represents the false data at different scales, and $ci$ represents the category information of the data sample.

Real data samples, generated fake data samples, and corresponding category information are used as the data of the model. Therefore, the discriminator in this study is different from discriminators in other models. The loss function corresponding to the discriminator in our approach:

$$L_{D_n} = -E_{x_n^r} log(D(x_n^r|ci)) - E_{x_n^f} log(1D(x_n^f|ci)) \tag{4}$$

where $x_n^r$ and $x_n^f$ respectively represent the data samples generated at different scales.

*2.6. Scene Classification*

Through the above operations, a large number of data sets with category information can be obtained. The obtained data set and the original data set can be combined to form a new data set, and more data sets can be obtained. Finally, we use the previously proposed Lie Group machine learning scene classification model and the typical scene classification model and adopt the obtained dataset to train and obtain the results.

### 3. Experimental Results and Analysis

*3.1. Experimental Datasets*

To verify the feasibility and validity of our approach, we chose the publicly available and challenging data sets, UC Merced [28] and AID [29]. The UC Merced dataset [28] contains 21 categories of scenes, and each category contains 100 images. The AID dataset [29] contains 30 categories of scenes, each of which has about 200 to 400 images. These two types of data sets contain a large number of scene categories and are representative, including (1)

high similarity between classes and diversity within classes; and (2) the scene being rich. The main characteristics of the above two datasets can be referred to in the literature [2,15].

### 3.2. Experiment Setup

To effectively simulate the lack of samples, we only select the number of sample data of each category as 20, 30, and 50, and the rest are used as test data samples. The generated samples are fused with the original data in a 1:1 manner, and these samples will be used in subsequent experiments.

On the basis of previous research [1] and repeated experiments, we set the experimental parameters, as shown in Table 2. In the actual algorithm design, the modules in our approach are optimized using the Adam optimizer. To verify the effectiveness of the model, we selected representative supervised generative models, such as SPG-GAN [14], OPGAN [21], NIGAN [9], sifting GAN [24] and unsupervised generative models, such as MARTA-GAN [22].

**Table 2.** Setting of experimental environment and other parameters.

| Item | Content |
|---|---|
| CPU | Inter Core i7-4700 CPU with 2.70 GHz $\times 12$ |
| Memory | 32 GB |
| Operating system | CentOS 7.8 64bit |
| Hard disk | 1T |
| GPU | Nvidia Titan-X $\times 2$ |
| Python | 3.7.2 |
| PyTorch | 1.4.0 |
| CUDA | 10.0 |
| Learning rate | $10^{-3}$ |
| Momentum | 0.7 |
| Weight decay | $5 \times 10^{-4}$ |
| Batch | 16 |
| Saturation | 1.5 |
| Subdivisions | 64 |

### 3.3. Results on the UC-Merced DataSet

### 3.3.1. Qualitative Analysis

The diversity and authenticity of fake samples generated by different models directly affect the accuracy of scene classification. Therefore, we compared fake data samples generated by different models, as shown in Figure 6.

In terms of global structure and local structure, our proposed approach has better global structure and local texture feature information. Especially for some scenes with the complex spatial distribution and geometric structure, the quality of samples generated by our approach is higher. The most important factor to improve the classification accuracy of the model is that the scene data samples need to contain rich physical structure features and texture feature information. In our method, we have adopted the object scale sample generation strategy from small-scale to large-scale and the residual connection operation, which can not only extract deeper feature information but also address the problem of model degradation. It can effectively generate fake data samples containing global structure and local details.

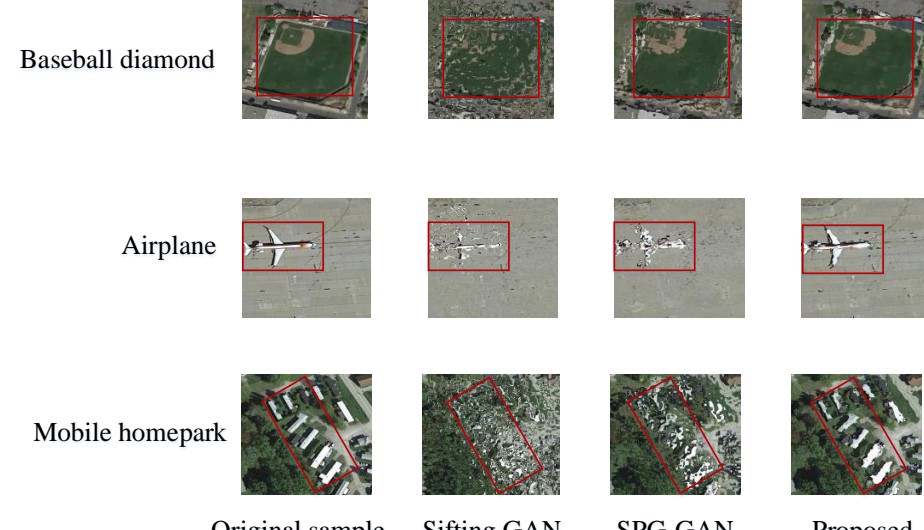

Figure 6. Comparison of samples generated by our approach and other approaches on the UC Merced dataset.

3.3.2. Quantitative Analysis

From Table 3, we found that the classification accuracy of these methods is relatively low. The main reason may be the lack of a large number of data samples containing category information. In the MARTA-GAN model, multiple feature layers are used to extract the feature information of the scene, and the traditional SVM classifier is used, which is also the reason for its low classification accuracy. When the number of samples is 20, our approach improves 4.38% compared with sifting GAN [24]. When the number of samples is 30, our approach increases 22.35% compared with MARTA-GAN [22]. To distinguish it from other models, the method we proposed in Table 3 is represented in bold.

**Table 3.** Classification accuracy on the UC Merced dataset.

| Models | Mode | OA(%) | | |
|---|---|---|---|---|
| | | 20 | 30 | 50 |
| Sifting GAN [24] | Supervised | 65.47 | 71.35 | 75.63 |
| SPG-GAN [14] | Supervised | 67.79 | 73.51 | 77.57 |
| OPGAN [21] | Supervised | 64.23 | 69.76 | 72.39 |
| NIGAN [9] | Supervised | 66.13 | 72.35 | 73.61 |
| MARTA-GAN [22] | Unsupervised | 42.97 | 53.22 | 59.86 |
| Proposed | Supervised | **69.85** | **75.57** | **79.74** |

When the number of samples in each category is 30, the experiment is carried out to obtain the confusion matrix as shown in Figure 7. Compared with other models, the classification accuracy of the MARTA-GAN model is lower, which is the same as the previous accuracy. Since the data samples containing category information are generated, the accuracy of some very easily confused categories (i.e., golf course, baseball diamond) is improved to a certain extent after adding the generated samples. Compared with the sifting GAN model because our proposed method can better preserve the texture structure and spatial structure, our method performs better in terms of accuracy.

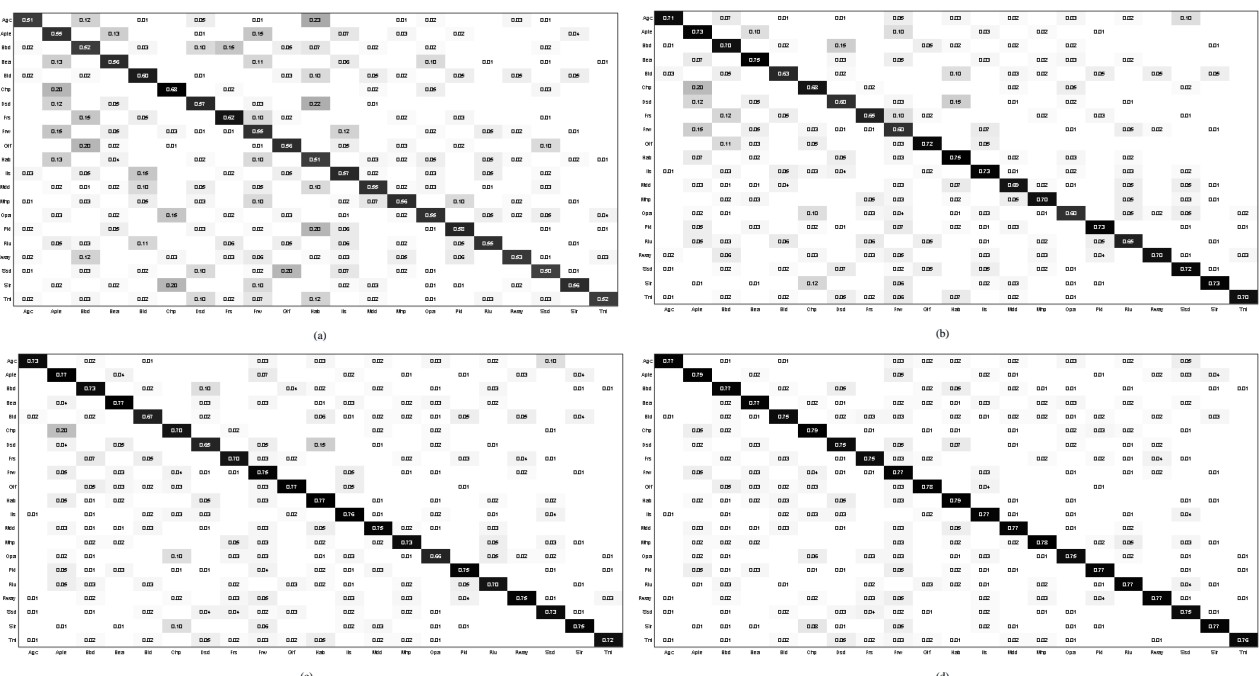

**Figure 7.** The confusion matrix of 30 sample training sets of UC Merced data set, (**a**) MARTA-GAN; (**b**) sifting GAN; (**c**) SPG-GAN; (**d**) proposed.

### 3.4. Results on AID DataSet

3.4.1. Qualitative Analysis

Figure 8 shows that some categories of data (such as commercial) are noisy in the fake data samples generated by the sifting GAN model. For uncomplicated data samples, Figure 8 shows that our method can obtain samples similar to the original data. For scenes with some features (such as rivers), sifting GAN is unable to generate clear boundary contours, and our approach achieves better results than sifting GAN. However, the boundary contour of the ground object can still be obtained. For structures with complex textures and geometric space structure of the scene, our method to generate fake data samples compared to other models generated fake data samples is better, mainly because our method adopts the object scale sample generation strategy. This strategy takes into account the semantic features of objects of different scales and adopts residual connection operation to effectively extract the feature information of the scene.

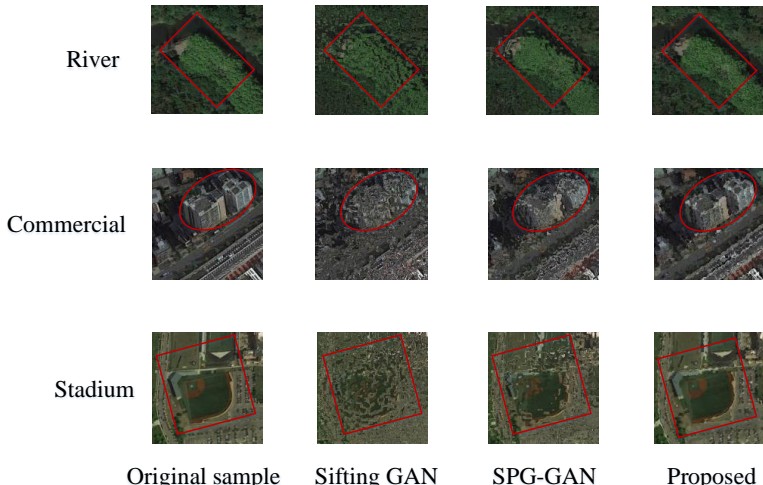

| River | | | |
| Commercial | | | |
| Stadium | | | |
| Original sample | Sifting GAN | SPG-GAN | Proposed |

**Figure 8.** Comparison of samples generated by our approach and other approaches on the AID dataset.

### 3.4.2. Quantitative Analysis

Table 4 lists the classification accuracy of different models in the AID dataset. When the number of each scene category is 20, 30, and 50, respectively, our approach achieves 49.17%, 59.79%, and 68.21%, which improves 3.66%, 2.72%, and 6.02% compared with the Sifting GAN [24] model. This experiment shows that our approach has achieved better results in different data samples. To distinguish it from other models, the method we proposed in Table 4 is represented in bold.

**Table 4.** Classification accuracy on the AID dataset.

| Models | Mode | OA(%) | | |
|---|---|---|---|---|
| | | 20 | 30 | 50 |
| Sifting GAN [24] | Supervised | 45.51 | 57.37 | 62.19 |
| SPG-GAN [14] | Supervised | 47.19 | 57.68 | 64.32 |
| OPGAN [21] | Supervised | 43.92 | 55.67 | 60.65 |
| NIGAN [9] | Supervised | 44.67 | 56.73 | 63.26 |
| MARTA-GAN [22] | Unsupervised | 31.42 | 45.69 | 51.35 |
| **Proposed** | Supervised | **49.17** | **59.79** | **68.21** |

When the number of scene categories is 50, the confusion matrix is shown in Figure 9. From Figure 9, we found that our method has higher accuracy than other models. For example, in the palace scene, the accuracy of the MARTA-GAN model is 55%, the accuracy of the SPG-GAN model is 67%, the accuracy of the sifting GAN model is 65%, and the accuracy of our method is 68%. The experimental results show that the data generated by our method effectively improve the accuracy of scene classification. Sifting GAN's accuracy is lower than our proposed model, which also shows that poor-quality data samples will affect the accuracy of scene classification. For complex scenes (such as dense residential, and sparse residential), the classification accuracy of sifting GAN, SPG-GAN, and our proposed are 60%, 65%, and 67%, respectively, which shows that our method still achieves the best classification accuracy. This is because we have comprehensively considered the external physical structure characteristics, internal socio-economic semantic characteristics, and high-level semantic characteristics of the scene. In addition, we can effectively ensure the diversity and authenticity of the generated data samples by effectively combining the above features with local spatial information.

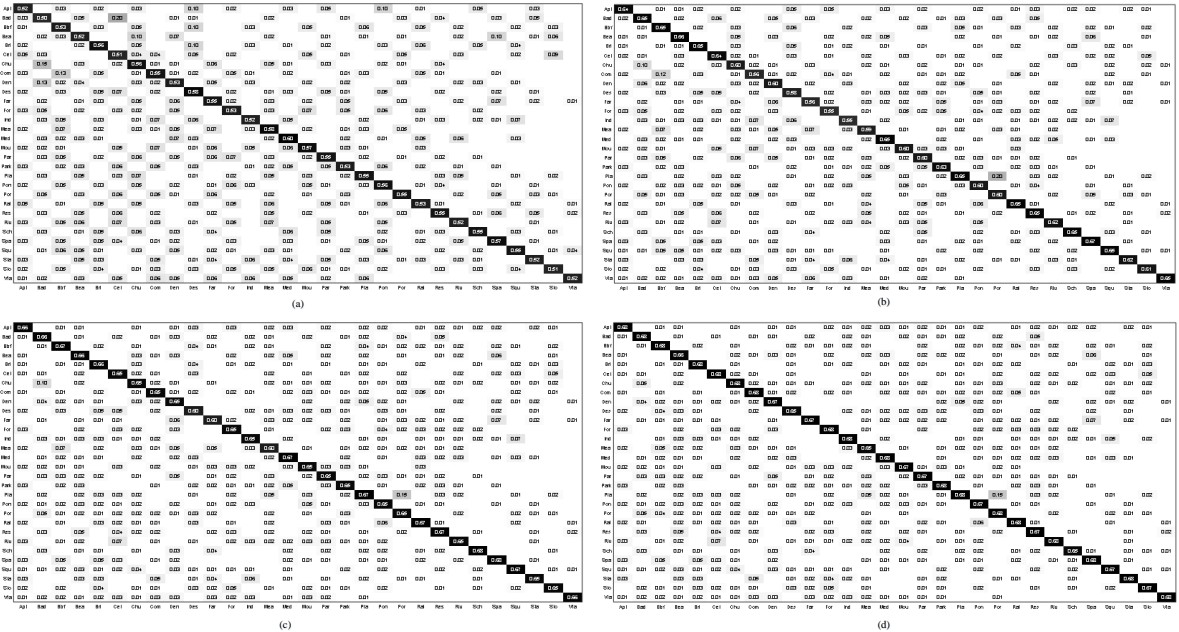

**Figure 9.** The confusion matrix of 50 sample training sets of AID data set, (**a**) MARTA-GAN; (**b**) sifting GAN; (**c**) SPG-GAN; (**d**) proposed.

## 4. Ablation Experiments

### 4.1. Model Parameters and Speed Analysis

Table 5 shows the parameters and running times of different models in the same experimental environment (namely, software environment and hardware environment). In terms of parameter size and running time, our approach has obvious advantages over other approaches. The main reason is that we introduce parallel dilated convolution and adopt the Lie Group region covariance feature matrix for representation, which effectively reduces the parameters of the model and reduces redundant features. In addition, the object-scale sample generation strategy loads parameters step by step, according to different scales, so our method has more advantages in running time. To distinguish it from other models, the method we proposed in Table 5 is represented in bold.

**Table 5.** Comparison of model parameter size and running time.

| Models | Mode | Parameters Size (M)↓ | Running Times (S)↓ |
|---|---|---|---|
| Sifting GAN [24] | Supervised | 36.4 | 1.218 |
| SPG-GAN [14] | Supervised | 31.9 | 2.080 |
| OPGAN [21] | Supervised | 23.0 | 1.165 |
| NIGAN [9] | Supervised | 36.1 | 1.207 |
| MARTA-GAN [22] | Unsupervised | 87.3 | 2.843 |
| Proposed | Supervised | **20.6** | **1.012** |

### 4.2. Influence of Internal Socio-Economic Semantic Features on Scene Classification

To verify the effect of internal socio-economic semantic features on classification accuracy, we conducted the following ablation experiments. In the experiment, we adopted the model without internal socio-economic semantic features and the model with internal socio-economic semantic features. The experimental results are shown in Table 6, which verifies the importance of including internal socio-economic semantic features. Further analysis shows that the number of parameters of the model does not increase significantly after adding the internal socio-economic semantic features, mainly because we adopt the Lie Group regional covariance feature matrix for representation. To distinguish it from other models, the method we proposed in Table 3 is represented in bold. To distinguish it from other modulars, the method we proposed in Table 6 is represented in bold.

**Table 6.** Influence of internal socio-economic semantic features.

| Modulars | OA (%) ↑ |
|---|---|
| Without internal socio-economic semantic features | 64.47 |
| Proposed | **68.21** |

### 4.3. Limitations of the Generated Samples

In the process of model operation, it is inevitable to generate failed data samples for further analysis, which is mainly the uncertainty of some complex scene categories and random noise sampling. Figure 10 shows some of the failed data samples, such as freeway and commercial. In the failed data samples, some do not have a clear boundary contour, and others do not have a complete category. Therefore, not all the data samples generated by the model are suitable for extending the original dataset, and failed data samples not only do nothing to help the accuracy of scene classification but also cause greater confusion, making it harder to distinguish scene categories.

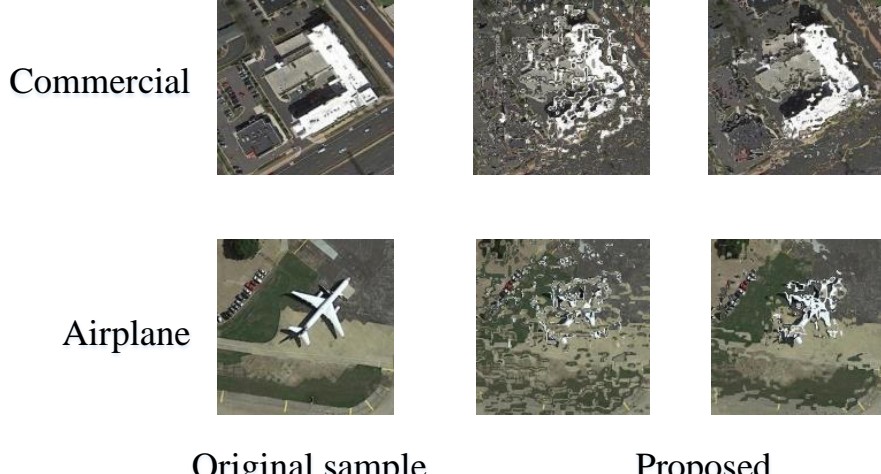

Commercial

Airplane

Original sample          Proposed

**Figure 10.** Sample of failed data generated by the model.

*4.4. Random Noise and Category Conditions*

We conduct ablation experiments on scene categories and random noise on the generated data samples, as shown in Figure 11. We divided into two cases: (1) lock random noise $rn$ and change category information $ci$; (2) change random noise $rn$ and lock category information $ci$. In the first case, rich category information and texture information can be generated. In the second case, data samples of the same category can be generated, which have different texture structures in some small details, and most data samples have high homogeneity.

From the above experiments, we found that, when $rn$ and $ci$ are included in the channel dimension, data samples containing rich category information can be obtained. Further analysis shows that category information plays a more important role. Since "randomness" and "condition" have their internal contradictions and limitations, the two cannot go together. To address this problem, we change the generation method of data samples, update the model constantly in the process of model training, and generate data samples of different scales synchronously.

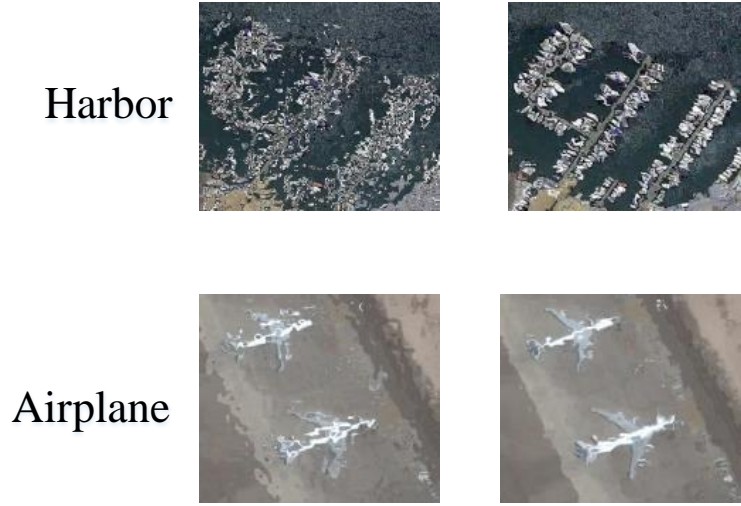

Harbor

Airplane

**Figure 11.** The samples generated by the model are affected by different random noise and category information. The first row represents the lock random noise $rn$ and the changed category information $ci$, and the second row represents the changed random noise $rn$ and the lock category information $rn$.

## 5. Conclusions

In this study, we have proposed an adversarial Lie Group feature learning network model. The biggest difference between this model and other models is that it can generate data samples containing category information. After fusing the generated fake data samples with the original data samples, a richer data sample set can be obtained. In the model, the object scale generation strategy is used, which can effectively generate data samples of different scales. In addition, in terms of feature learning, we optimized on the basis of previous research, supplemented the internal socio-economic semantic features of scenes, and further enhanced the representation ability of scenes.

Since there are still some gaps between the generated data samples and the original data samples, in the future, we will continue to deeply study Lie Group and optimize the approach to generate more realistic data samples.

**Author Contributions:** Conceptualization, C.X.; G.Z. and J.S.; methodology, C.X. and J.S.; software, J.S.; validation, C.X. and G.Z.; formal analysis, J.S.; investigation, C.X.; resources, C.X. and G.Z.; data curation, J.S.; writing—original draft preparation, C.X.; writing—review and editing, C.X.; visualization, J.S.; supervision, J.S.; project administration, J.S.; funding acquisition, C.X. All authors have read and agreed to the published version of the manuscript.

**Funding:** This work was partially supported by the National Natural Science Foundation of China (Research on Urban Land-use Scene Classification Based on Lie Group Spatial Learning and Heterogeneous Feature Modeling of Multi-source Data), under Grant No. 42261068.

**Institutional Review Board Statement:** Not applicable.

**Informed Consent Statement:** Not applicable.

**Data Availability Statement:** Data associated with this research are available online. The UC Merced dataset is available for download at http://weegee.vision.ucmerced.edu/datasets/landuse.html (accessed on 12 November 2021). AID dataset is available for download at https://captain-whu.github.io/AID/ (accessed on 15 December 2021).

**Acknowledgments:** The authors would like to thank four anonymous reviewers for carefully reviewing this study and giving valuable comments to improve the study.

**Conflicts of Interest:** The authors declare no conflict of interest.

## Abbreviations

| | |
|---|---|
| AID | Aerial Image Dataset |
| BN | Batch Normalization |
| CGAN | Conditional Generative Adversarial Network |
| CNN | Convolutional Neural Network |
| DCA | Discriminant Correlation Analysis |
| GAN | Generative Adversarial Network |
| GCH | Global Color Histogram |
| HRRSI | High-Resolution Remote Sensing Images |
| OA | Overall Accuracy |
| ReLU | Rectified Linear Units |
| SeLU | Scaled Exponential Linear Units |

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
