# Peer review of "Adversarial Remote Sensing Scene Classification Based on Lie Group Feature Learning"

_remotesensing, doi:10.3390/rs15040914_

Round 1
Reviewer 1 Report
This manuscript is mainly designed for the model that most of the existing models cannot generate samples containing scene category information. This model uses Lie Group feature learning. In the algorithm design, the internal socio-economic semantic features are supplemented, effectively increasing the feature representation capability of the scene. The model also adopts a supervised generation mode, which takes the sample category information and samples together as the input of the model, optimizes the category information generation strategy in the loss function, and uses the object-scale sample generation strategy to generate data samples of different scales. In addition, the eight neighborhood method is also used to enhance the correlation between ground objects. The experiments in the manuscript are sufficient, the experimental results are convincing, and the ablation experiment has been carried out. The logic is very clear. I think this manuscript is acceptable. In addition, I suggest to the author that unsupervised methods should be considered in future studies to further expand the scope of learning.
Supplementary comment:
1. This manuscript is mainly aimed at the problem of a lack of data samples in the field of remote sensing, especially since the existing GAN cannot generate data samples containing scene category information. It proposed a novel supervised adversarial Lie Group feature learning network model, which combines Lie Group machine learning and GAN, and adopted the object scale sample generation strategy so that the generated data sample contains richer feature information.
2. The research content (topic) in this manuscript belongs to the original research. The main reasons are as follows: (1) The combination of Lie Group machine learning and GAN improved the interpretability and comprehensibility of the model from the perspective of Lie Group machine learning, which is essentially different from previous research, and is also the highlight and innovation of this manuscript; (2) The generator in the model adopted parallel dilated convolution, and compared it with ordinary convolution. It used the form of a table, which is very clear and easy for readers to understand; (3) The activation functions in the model are also compared, and the reason why the SeLU function is used in the manuscript is compared and analyzed; (4) The author designed and developed a probability enhancement strategy based on neighborhood correlation to further improve the accuracy of sample scenarios. In the manuscript, the author compared and analyzed four neighborhoods and eight neighborhoods, explains why eight neighborhoods are selected, and the details in the manuscript are considered well; (5) The experimental part in the manuscript is very detailed and full, and the ablation experiment is taken into account, and the advantages of the model are explained through comparative analysis. In summary, this manuscript is very suitable for publication in this journal.
3. Compared with the published materials (articles), the main contributions are as follows: (1) The integration of Lie Group machine learning into GAN improves the interpretability and comprehensibility of the model from the perspective of Lie Group machine learning, which is an innovation and highlight, and provides a new solution for the sample generation method; (2) The activation function, parallel dilated convolution, object-scale sample generation strategy and probability enhancement strategy based on neighborhood correlation in the manuscript, etc. Compared with the existing models, the model proposed by the author has certain innovations, reduces the characteristic dimension of the model, improves the performance of the model, and the generated data samples contain more information; (3) The experimental part in the manuscript is detailed and the experimental analysis is accurate, which better explains the function of each module of the model.
4. This manuscript is relatively good on the whole, with clear ideas and sufficient experiments. It is suggested to check the format of the manuscript (such as references).
5. The author carried out a detailed experiment in the experimental part, fully analyzed and discussed, and the conclusion was consistent with the evidence and argument proposed, which solved the main problems posed.
6. The references in the manuscript are closely related to the content, and the references cited are also representative and relatively new. In the comparative experiment, the cited literature is also very representative, which can reflect the superiority of the model proposed by the manuscript.
7. The tables and pictures in the manuscript are very clear, which is helpful for readers to read and understand, and can clearly express the meaning.
Reviewer 3 Report
This paper introduced a GAN-based network for generating samples with categorical labels to augment limited training labelled samples in a remote sensing scene classification task. The author adopted supervised approach with Lie Group feature learning and multi-scale sample strategy to train the network, and generate samples of different resolutions.
The experimental results on UC-Merced and AID datasets looks great , which makes the proposed method a promising to address limited label data problem in remote sensing scene classification.
The paper is well written, easy to read and comprehend. However, there are some minor issues (as stated below) that I would like the authors to consider and address.
(1) The images in Figures 6, 8, 10 and 11 should be enlarge a bit to improve visual clarity.
(2) The values in the confusion matrices in Figure 7 and 9 are not readable. I suggest the authors should enlarge them and change the font size and font style in the confusion matrices to make the values readable.
(3) I suggest the authors should bold the best results in the Tables 3, 4, 5, and 6 so that readers can identify the best results easily.
(4) There is a typo error in "...which improves 3.66%, 2.72%, and [60.2%] compared with the SiftingGAN..." in Section 3.4.2. The 60.2% --> 6.02%
Reviewer 4 Report
A novel supervised adversarial Lie Group feature learning network is proposed. In the case of limited data samples, the model can effectively generate data samples with category information. There are two main differences between our method and the traditional GAN. First, our model takes category information and data samples as the input of the model and optimizes the constraint of category information in the loss function, so that data samples containing category information can be generated. Secondly, the object scale sample generation strategy is introduced, which can generate data samples of different scales and ensure that the generated data samples contain richer feature information. This paper explains the theory in more detail and the data are clearly listed. This is an interesting research paper. There are some suggestions for revision.
1. The motivation is not clear. Please specify the importance of the proposed solution.
2. The listed contributions are a little bit weak. Please highlight the innovations of the proposed solution.
3. The related work is not detailed enough, and the description of remote sensing images is too little. In addition, for the development of the research task of this paper, we can further refer to more literature to enhance the integrity of the article.
4. In the introduction of the method in this paper, the introduction of each module is not detailed enough. Using three-dimensional pictures to represent the composition and function of each module will be clearer.
5. Please discuss how to obtain the suitable parameter values used in the proposed solution.
6. Using appropriate formulas, especially for the focus and innovation of the article, will highlight the logical rigor of the article. Why some of the key parts are a stroke, neither the formula nor highlight the focus.
7. In the experimental part, there is not much introduction to the datasets. Please supplement the reasons for selecting these two data sets and the introduction of the basic information of the datasets.
8. In the comparison with other methods, whether several methods can be added to verify the effectiveness of the method.
9. In the method comparison, the advantages of this method and the shortcomings of other methods are not introduced in detail. Marking and sketching in the picture will be clear at a glance.
10. The number of references is insufficient, it will be better to search and cite more latest research, which can better reflect the innovations of this manuscript.
11. The experimental results are not convincing. Please compare the proposed solution with more recently published solutions.
Round 2
Reviewer 4 Report
All my concerns have been addressed. I recommend this paper for publication.